# Sustainable Service-Learning through Massive Open Online Courses

**Dominik E. Froehlich** 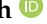

Centre for Teacher Education, University of Vienna, 1090 Vienna, Austria; dominik.froehlich@univie.ac.at

**Abstract:** In this opinion article, I discuss the role of online learning, especially Massive Open Online Courses (MOOCs), in making what I call "resource-intensive pedagogies" more sustainable. Specifically, I zero in on the framework of service-learning. I present a case study of a long-standing and successful service-learning project in the context of teacher education. This case-based evidence suggests that MOOCs can be highly instrumental in individualizing course content and increasing the course concept's scalability. The article ends by suggesting three avenues for further research to explore this nexus of online learning and resource-intensive pedagogies.

**Keywords:** distance learning; online learning; service-learning

## 1. Introduction

The past decade has seen a profound transformation in the educational landscape. With the advent of technologies such as online platforms, video conferencing, and virtual reality, distance education is at an exciting crossroads, merging cutting-edge technology with evolving pedagogical techniques [1]. This convergence is molding the future of learning, teaching, and knowledge exchange, with its impact reverberating beyond academia, into the wider society.

But as we navigate this transformation, we are confronted with a stark reality. Our comprehension and implementation of distance education often remain trapped within a narrow didactic framework [2,3]. Traditional interpretations of distance education have been primarily unilateral, focusing mainly on information delivery from educators to students; for example, within the inverted learning framework [4,5]. This teacher-centered model, while practical, often overlooks the multiplicity of pedagogical strategies that can lead to more interactive, enriching, and holistic learning experiences [6]. In essence, the conventional perspective has been myopically centered around content delivery, ignoring the potential for interaction, collaboration, and experiential learning. This approach fails to take advantage of the many opportunities that technology offers for transforming pedagogical practice, consequently limiting the potential of distance education to engage learners and foster deeper understanding.

In this context, and in the spirit of "flexible pedagogies" [7], it becomes crucial to reassess and expand our pedagogical outlook to fully harness the potential of distance education. By integrating a broader array of pedagogical perspectives not native to the digital domain, we can stimulate creativity and innovation in digital and hybrid learning environments and their sustainability. These pedagogical shifts can also inspire novel research pathways in education. By exploring how different teaching methodologies interact with digital platforms, we can build a more comprehensive understanding of how learning occurs in these environments. This knowledge can further guide the design of digital learning experiences that cater to diverse learning styles and needs.

As we aspire to harness the full potential of technology within the educational realm, it becomes imperative to transcend the confines of traditional teaching models, which encompasses traditional pedagogies that have often been adopted in the context of distance

education. Conventionally, such models have been disproportionately centered on content delivery, treating distance education as a simplistic, unidirectional conduit for information transmission [8]. In an era characterized by unprecedented technological advancements, there is a pressing need to re-envision distance education as more than just a passive medium. Rather, it can be reconceptualized as a dynamic, interactive platform capable of engendering profound, collaborative learning experiences. The promise of distance education extends beyond the provision of access to learning materials; it offers potential solutions for making (any?) pedagogical practice more sustainable and less resource-intensive.

Resource-intensive pedagogies often necessitate considerable financial and infrastructural support. The strategic incorporation of distance education can provide avenues for circumventing these challenges. By leveraging digital technologies, we can deliver quality education with substantially lower consumption of physical resources, making these pedagogical practices more economically viable and environmentally sustainable. This capacity of distance education to enhance the sustainability of educational processes aligns with the increasingly critical global agenda of sustainable development. Therefore, as we navigate the future of education, it is crucial that our pedagogical strategies align with the changing landscape. By widening our pedagogical perspectives, we can ensure that distance education evolves in sync with the demands of our rapidly advancing world, thereby equipping learners with the requisite skills and knowledge to flourish in the digital age. Furthermore, it helps us address the crucial requirement of sustainability in education.

## 2. Sustainable Service-Learning: Distance Education Strategies to Support Resource Intensive Pedagogies

### 2.1. Background of Service-Learning

In our quest to broaden pedagogical perspectives within distance education, an exemplary path lies in embracing service-learning. Regarded as a transformative educational practice, service-learning seamlessly integrates meaningful community service with instruction, intertwining it with reflective thinking to cultivate an enriched, well-rounded learning experience [9,10]. This pedagogical approach not only enhances academic knowledge but also advocates for civic responsibility and strengthens communities [11].

Service-learning offers a dynamic counterpoint to the conventional, didactic, content-focused model that currently typifies distance education. Rather than being solely a consumer of knowledge, the learner becomes an active participant in knowledge creation [12]. This shift to active participation provides learners with opportunities to apply and test theoretical concepts in real-world scenarios. Thus, learning evolves from a static process of information acquisition to a dynamic process of knowledge construction, culminating in deep and lasting understanding. In essence, service-learning transcends the traditional classroom boundaries and redefines the concept of an educational environment. The community transforms into an active learning space, acting as an extension of the classroom. This expanded classroom fosters the creation and application of knowledge in real-world settings, breathing life into theoretical concepts and forging a strong connection between academic learning and societal needs [13].

Service-learning cultivates an experiential form of education that values the application of knowledge to real-life situations. It encourages learners to step outside the familiarity of their comfort zones, prompting them to grapple with complex, real-world problems. This form of education acknowledges and champions the merits of learning through doing and reflecting, thereby fostering an educational system that appreciates and values experiential learning [14]. Furthermore, service-learning underscores the importance of civic engagement and social responsibility in education. It cultivates an understanding of how academic knowledge can be employed to address societal challenges and contribute to the betterment of communities. Consequently, learners develop a sense of responsibility and empathy towards their communities, and understand the integral role they can play in society's progress.

In sum, integrating service-learning within distance education presents an opportunity to move beyond a narrow didactic lens and also focus on aspects of sustainability of teaching practices. By emphasizing the application of knowledge, fostering civic engagement, and nurturing personal development, service-learning can reshape distance education into a more interactive, enriching, and meaningful endeavor. It serves as a testament to the potential of broadening our pedagogical horizons in enhancing the quality and relevance of distance education in our fast-evolving world.

### 2.2. The Challenge of Service-Learning: Resource Intensity

One of the potential challenges of service-learning is its resource-intensiveness [15]. Implementing such pedagogy may require significant logistical coordination, time investment, and institutional support. However, it is here that the strategic integration of distance education can serve as a potential remedy. Distance education strategies, with their inherent scalability and flexibility, can provide mechanisms for alleviating some of the resource constraints associated with service-learning.

For instance, digital platforms can be harnessed to coordinate service-learning activities, reducing the need for extensive physical infrastructure and administrative support. Online communication tools can facilitate interactions between students, instructors, and community partners, thereby overcoming geographical barriers and enhancing collaborative efficiency. Furthermore, the availability of a plethora of online learning resources can significantly diminish the financial burden of acquiring learning materials, making the pedagogy more accessible. Moreover, distance education can offer greater temporal flexibility, enabling learners to engage with service-learning activities at their convenience. This aspect is particularly beneficial for working adults or those with familial responsibilities, for whom traditional, time-bound educational models may prove challenging. Thus, integrating distance education strategies within service-learning can make this transformative pedagogy more sustainable, scalable, and inclusive, bringing it within the reach of a broader learner demographic.

In effect, the incorporation of distance education strategies within service-learning can serve as a powerful mechanism to increase the sustainability of this pedagogical approach. It paves the way towards making service-learning, an otherwise resource-intensive pedagogy, more manageable, accessible, and sustainable, thereby expanding its reach and potential impact. This integration can serve as an exemplar for how we can leverage technology to foster innovative, immersive, and sustainable educational experiences.

## 3. Case Study

### 3.1. Background and Context

In this section, I provide a short case study about an instance of service-learning that was significantly leveraged through the means of distance education, in general, and Massive Open Online Courses (MOOCs), in particular: the Teaching Clinic (https://teachingclinic.org/, accessed 1 July 2023). The Teaching Clinic is a pioneering educational initiative based in two Austrian higher teacher education programs, and presents a compelling illustration of how service-learning can be integrated into distance education [16]. As an exemplar of innovative pedagogical practice, the Teaching Clinic seamlessly weaves together the strengths of MOOCs, and also Small Private Online Courses (SPOCs) [17] and service-learning, thereby facilitating rich, multifaceted learning experiences. This dynamic fusion allows students to engage in an array of research projects directly proposed by teachers, covering an extensive spectrum of topics that range from political education against racism to the non-digital teaching of digital literacy [16].

The Teaching Clinic can be summarized in three steps. First, in-service teachers submit individual challenges via an online form to the Teaching Clinic. This is based on an open call, which may include concrete challenges in a classroom, exploratory projects to try out something new, or anything in between. Second, the challenges are presented to pre-service

teachers who participate in a Teaching Clinic course. They then join the project that meets their interests and learning goals best. Third, the pre-service teachers collaborate with the in-service teachers to tackle the project. Specifically, design-based research [18] is applied to find scientific solutions to the posed challenges.

In the unique model of the Teaching Clinic, each student's learning path is highly individualized, and meticulously designed to meet the specific demands of each unique project. This tailoring of the learning experience affords students a remarkable level of flexibility and autonomy in their educational journey. It also promotes a more learner-centric pedagogical approach, allowing students to feel more connected to their learning, thereby enhancing motivation and engagement [19,20]. Moreover, the Teaching Clinic brings to the forefront the immense potential of MOOCs within a service-learning framework. The access to an extensive catalogue of MOOCs, tailored to align with students' individual learning objectives, fosters a highly personalized and engaging learning experience. This personalization empowers students to take charge of their learning journey, fostering a sense of self-direction and independence that is critical in fostering lifelong learning habits [21].

From the perspective of instructors, the integration of MOOCs with service-learning presents unprecedented opportunities for transformative teaching practices. With the bulk of instructional content delivered through MOOCs, instructors can transition from their traditional roles of information delivery to becoming facilitators and guides. This shift allows them to allocate more time and resources towards coaching, providing individualized support, and supervising a variety of projects concurrently. This shift in teaching practice, termed as 'mass-individualization', points towards a promising trajectory for the future of higher education [22–25]. It envisions an educational landscape where each student's learning journey is tailored to their unique needs and interests, thereby enhancing learning outcomes and student satisfaction.

### 3.2. Leverage through Distance Learning

The marriage of MOOCs and service-learning not only promises to transform student learning experiences, but also brings significant benefits on an institutional level. By incorporating service-learning within the framework of distance education, educational institutions can substantially broaden their reach. This facilitates the creation of more project partnerships with various stakeholders, resulting in the expansion of community outreach activities and fostering stronger ties between the institution and its surrounding communities. The integration of MOOCs and service-learning offers a fascinating juxtaposition of global and local perspectives. The former, with its boundless reach, brings to the fore an array of global viewpoints and knowledge, while the latter, with its emphasis on addressing specific community-based issues, provides a local lens through which to understand and apply this knowledge. This unique interplay has the potential to stimulate cross-institutional collaborations, fostering an interconnected, global educational community that transcends geographical boundaries. Moreover, the creators of MOOCs also stand to reap substantial benefits from this innovative integration. By closely analyzing how students engage with their courses within the context of specific service-learning projects, MOOC creators can gain a more detailed understanding of course utilization. These insights, rooted in real-world applications and practices, can help creators refine their courses, making them more responsive to the evolving needs of students and different institutional contexts.

The practical implementation of service-learning, as evident in the case of the Teaching Clinic, demonstrates the sustainability and scalability achieved through the strategic integration of distance education methodologies. The use of MOOCs as a primary mode of content delivery has drastically amplified the reach of the program, facilitating national scaling that previously seemed impractical with traditional, localized service-learning models. The modularity and flexibility offered by MOOCs have equipped the Teaching Clinic with the ability to customize learning paths for individual students, effectively addressing

their unique learning objectives and project requirements. This flexibility not only makes the learning process more engaging and effective, but also greatly reduces resource strain on the institution by enabling a higher level of self-guided learning.

On a broader level, the Teaching Clinic's successful national scaling is a promising indicator of the potential of this model for international expansion. The adaptability and scalability inherent in MOOCs, combined with the transformative potential of service-learning, set the stage for a truly global educational community. The internationalization of the Teaching Clinic would enable a greater diversity of service-learning projects, facilitating cross-cultural collaborations and enriching the learning experience with diverse global perspectives.

By creating an educational model that is both scalable and sustainable, the Teaching Clinic serves as an exemplar of how distance education can transform resource-intensive pedagogies like service-learning. Its success is a testament to the possibilities that lie at the intersection of technology and pedagogy, promising a future of education that is not just broad and inclusive, but also environmentally and institutionally sustainable.

## 4. Venturing beyond the Crossroads of Service-Learning and Distance Education: An Agenda for Further Research

We are at the precipice of an exciting era, where the intersection between these two pedagogical realms holds the promise of revolutionizing the way we perceive and experience education. However, to navigate this new terrain, numerous unanswered questions and challenges must be addressed. Here, I outline three of them, related to the integration of MOOCs within service-learning, the (digitally enhanced?) network aspects of service-learning, and the research methods to study the field.

The first point concerns the question of how we may refine the integration of MOOCs within service-learning (and other resource-intensive pedagogical) environments to optimize their potential. So far, I have provided case-based evidence of how this integration has fruitfully worked in a single instance (that, however, has been repeated in different institutions and over time). Indeed, this implementation of the Teaching Clinic can serve as a point of leverage for further large-scale implementation [26]. Nevertheless, we know little about other ways of integrations, which calls for further experimentation in higher education seminars and thorough evaluation. This could range from employing sophisticated learning analytics to personalize the learning experience, to utilizing virtual reality to simulate real-world service situations, or even fostering online communities for more robust reflection and collaboration. These digital tools could further enrich the service-learning experience, providing learners with a more immersive, engaging, and effective learning journey.

A second research avenue leads to the networks of stakeholders that are specific to service-learning. Given that service-learning involves external partners, one may also question how technology may be used to make this aspect more sustainable. What different configurations might these partnerships take to suit diverse contexts and learner needs [27]? Policymakers, administrators, and educators need to deliberate on how to adapt existing structures and systems to facilitate this innovative blend of learning modalities. There is a need for professional development programs for educators to equip them with the skills and knowledge necessary to navigate this new pedagogical landscape effectively.

Lastly, and related to research, future research should also probe deeper into the impacts of such integration on learner outcomes, including both content-related and wider aspects of digital competencies. There is a need to understand the full extent of how service-learning within MOOCs impacts students' academic performance, personal growth, and civic responsibility. While longitudinal studies and meta-analyses could provide empirical evidence on the long-term benefits and potential challenges of this pedagogical fusion, it is especially mixed methods research that is needed to add qualitative understanding to the statistics [28] and to understand the complex interplay of learning processes happening on the individual level, the level of the seminar, or the wider network provided by the

service-learning stakeholders or through the participation in the MOOC [29]. This may also stimulate a re-evaluation of the effectiveness of MOOCs in general, because while current research has identified a host of challenges with MOOCs [30,31], this might be (as I have argued in this article) due to a too narrow didactical framing in the first place.

**Funding:** This research received no external funding.

**Institutional Review Board Statement:** Not applicable.

**Informed Consent Statement:** Not applicable.

**Data Availability Statement:** Not applicable.

**Acknowledgments:** Open Access Funding by the University of Vienna.

**Conflicts of Interest:** The author declares no conflict of interest.

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
