# Peer review of "Sustainable Service-Learning through Massive Open Online Courses"

_sustainability, doi:10.3390/su151813522_

Round 1

Reviewer 1 Report

In Sustainable Service-Learning through MOOCs, the author describes the role of MOOCS as a means to sustainability for resource-intensive pedagogies. I like that the author frames the introduction that people should push their thinking to flexible pedagogies and with existing resources already available like MOOCs. The author provides an example of how a MOOC plays a role in service learning. I don't have a major concern or suggestion. The author does self-cite a good bit, but in the essay format, that makes sense. 

The conclusions that the author makes for further research are in line with the argument the author is making to consider MOOCs for flexible pedagogy, and that call for action would hopefully inspire others to take up these research questions for future work. This work adds a unique perspective to the body of work in that the essay focuses on the use of MOOCs in an argument for the sustainability of pedagogical resources that I have not seen before. One possible addition would be to contrast this view with a reference making the argument that MOOCs are not important anymore.

Author Response

> In Sustainable Service-Learning through MOOCs, the author describes the role of MOOCS as a means to sustainability for resource-intensive pedagogies. I like that the author frames the introduction that people should push their thinking to flexible pedagogies and with existing resources already available like MOOCs. The author provides an example of how a MOOC plays a role in service learning. I don't have a major concern or suggestion. The author does self-cite a good bit, but in the essay format, that makes sense.

Thank you. Yes, the essay is largely based on previous case studies that inform and substantiate the argument, hence these specific references.

> The conclusions that the author makes for further research are in line with the argument the author is making to consider MOOCs for flexible pedagogy, and that call for action would hopefully inspire others to take up these research questions for future work. This work adds a unique perspective to the body of work in that the essay focuses on the use of MOOCs in an argument for the sustainability of pedagogical resources that I have not seen before.

Thank you for the validation!

> One possible addition would be to contrast this view with a reference making the argument that MOOCs are not important anymore.

Thank you, this is a very interesting point. I have taken up your suggestion and have added a sentence in the discussion section (pointing to a re-evaluation of MOOC-effectiveness with a wider didactical frame in the background):

This may also stimulate a re-evaluation of the effectiveness of MOOCs in general; because while current research has identified a host of challenges with MOOCs [34,35], this might be (as I have argued in this article) due to a too narrow didactical framing in the first place.

Reviewer 2 Report

The author has produced an interesting paper on the development of a ‘flexible pedagogy’ to enhance the delivery of education and learning by the application of theory to practice, thus developing the students understanding of the topic.  The aim of the paper is to explore the intersection of online learning and resource intensive pedagogies.  The exploration discusses the role of MOOC’s to implement the concept of ‘resource-intensive pedagogies’ which are a departure from the traditional online distance education pedagogies by utilising current available technologies to provide an improved learning resource to distance education is innovative, especially for students involved in the service industries.  Such students can experience difficulties with the timing of current learning activities which may be alleviated with the use of AI enhanced resources allowing for the immediate feedback to student queries.   Historically, distance education has been restrained by the technology available for the transfer of theoretical knowledge with practical examples which may not have been relevant to the students’ needs by ensuring access to learning resources.

The author proposes further pedagogical research to explore the current distance learning resources be undertaken to construct a database from which new, and increasingly relevant, learning resources may be developed to replace the existing distance learning resources.  As part of this exploration, the author has included a case study of an online course to improve existing instructional resources which implement the integration of instructional design and online resource development such as that developed by the Teaching Clinic to provide an interactive learning experience that would ensure the involvement of the learner.  The author advocates the inclusion of distance education strategies within service-learning to improve sustainability, scalability, and inclusiveness to provide greater access across demographics.  Lastly, the author proposes three areas for further research and development: (1) the integration of MOOC’s into service-learning (2) the adaption of existing structures and systems into the pedagogy by professional development programs, and (3) research into the outcomes of the pedagogy into the student outcomes.

The discussion presented outlines the opinions of the author to the development, implementation, and outcomes of the proposed ‘flexible pedagogy’.  These arguments present a compelling case for the benefits of the inclusion of technology in instructional design and production of higher quality learning resources to assist in the development of understanding by the learner of the topic being learnt.  The paper is supported by the reference which are relevant to the opinions of the author. Overall, this opinion paper provides an explanation of the ‘flexible pedagogy’, an innovative learning process for distance education students, teachers, and administrators.  The provision in greater detail of the pedagogy in the case study would assist in the understanding the authors’ opinions.  The references are appropriate to the paper and indicate the direction of the authors’ opinions.

Author Response

> The author has produced an interesting paper on the development of a ‘flexible pedagogy’ to enhance the delivery of education and learning by the application of theory to practice, thus developing the students understanding of the topic.  The aim of the paper is to explore the intersection of online learning and resource intensive pedagogies.  The exploration discusses the role of MOOC’s to implement the concept of ‘resource-intensive pedagogies’ which are a departure from the traditional online distance education pedagogies by utilising current available technologies to provide an improved learning resource to distance education is innovative, especially for students involved in the service industries.  Such students can experience difficulties with the timing of current learning activities which may be alleviated with the use of AI enhanced resources allowing for the immediate feedback to student queries.   Historically, distance education has been restrained by the technology available for the transfer of theoretical knowledge with practical examples which may not have been relevant to the students’ needs by ensuring access to learning resources.

The author proposes further pedagogical research to explore the current distance learning resources be undertaken to construct a database from which new, and increasingly relevant, learning resources may be developed to replace the existing distance learning resources.  As part of this exploration, the author has included a case study of an online course to improve existing instructional resources which implement the integration of instructional design and online resource development such as that developed by the Teaching Clinic to provide an interactive learning experience that would ensure the involvement of the learner.  The author advocates the inclusion of distance education strategies within service-learning to improve sustainability, scalability, and inclusiveness to provide greater access across demographics.  Lastly, the author proposes three areas for further research and development: (1) the integration of MOOC’s into service-learning (2) the adaption of existing structures and systems into the pedagogy by professional development programs, and (3) research into the outcomes of the pedagogy into the student outcomes.

Thank you for this concise summary! One very minor clarification: The (service-learning) course itself is not an online course, but it is highly supported/leveraged through MOOCs.

> The discussion presented outlines the opinions of the author to the development, implementation, and outcomes of the proposed ‘flexible pedagogy’.  These arguments present a compelling case for the benefits of the inclusion of technology in instructional design and production of higher quality learning resources to assist in the development of understanding by the learner of the topic being learnt.  The paper is supported by the reference which are relevant to the opinions of the author. Overall, this opinion paper provides an explanation of the ‘flexible pedagogy’, an innovative learning process for distance education students, teachers, and administrators.

Thank you for your assessment of the overall argument!

> The provision in greater detail of the pedagogy in the case study would assist in the understanding the authors’ opinions. 

This is a good point; on re-reading the essay I indeed found some basic information about the Teaching Clinic to be lacking. I have therefore added a full paragraph to outline the concept. More information can be found at the given references and the specified URL.

This is what I added:

The Teaching Clinic can be summarized in three steps. First, in-service teachers sub-mit individual challenges via an online form to the Teaching Clinic. This is based on an open call which may include concrete challenges in a classroom, exploratory projects to try out something new, or anything in between. Second, the challenges are presented to pre-service teachers who participate in a Teaching Clinic course. They then join the project that meets their interests and learning goals best. Third, the pre-service teachers collabo-rate with the in-service teachers to tackle the project. Specifically, design-based research [21] is applied to find scientific solutions to the posed challenges.

> The references are appropriate to the paper and indicate the direction of the authors’ opinions.

Thank you.
